# CryoFormer: Continuous Heterogeneous Cryo-EM Reconstruction using Transformer-based Neural Representations

## Abstract

Cryo-electron microscopy (cryo-EM) allows for the high-resolution reconstruction of 3D structures of proteins and other biomolecules. Successful reconstruction of both shape and movement greatly helps understand the fundamental processes of life. However, it is still challenging to reconstruct the continuous motions of 3D structures from hundreds of thousands of noisy and randomly oriented 2D cryo-EM images. Recent advancements use Fourier domain coordinate-based neural networks to continuously model 3D conformations, yet they often struggle to capture local flexible regions accurately. We propose *CryoFormer*, a new approach for continuous heterogeneous cryo-EM reconstruction. Our approach leverages an implicit feature volume directly in the real domain as the 3D representation. We further introduce a novel query-based deformation transformer decoder to improve the reconstruction quality. Our approach is capable of refining pre-computed pose estimations and locating flexible regions. In experiments, our method outperforms current approaches on three public datasets (1 synthetic and 2 experimental) and a new synthetic dataset of PEDV spike protein. The code and new synthetic dataset will be released for better reproducibility of our results.

## 1 Introduction

Dynamic objects as giant as planets and as minute as proteins constitute our physical world and produce nearly infinite possibilities of life forms. Their 3D shape, appearance, and movements reflect the fundamental law of nature. Conventional computer vision techniques combine specialized imaging apparatus such as dome or camera arrays with tailored reconstruction algorithms (SfM (Schonberger & Frahm, 2016) and most recently NeRF (Mildenhall et al., 2020)) to capture and model the fine-grained 3D dynamic entities at an object level. Similar approaches have been adopted to recover shape and motion at a micro-scale level. In particular, to computationally determine protein structures, cryo-electron microscopy (cryo-EM) flash-freezes a purified solution that has hundreds of thousands of particles of the target protein in a thin layer of vitreous ice. In a cryo-EM experiment, an electron gun generates a high-energy electron beam that interacts with the sample, and a detector captures scattered electrons during a brief duration, resulting in a 2D projection image that contains many particles. Given projection images, the single particle analysis (SPA) technique iteratively optimizes for recovering a high-resolution 3D protein structure (Kühlbrandt, 2014; Nogales, 2016; Renaud et al., 2018). Applications are numerous, ranging from revealing virus fundamental processes (Yao et al., 2020) in biodynamics to unveiling drug-protein interactions (Hua et al., 2020) in drug development.

Compared with conventional shape reconstruction of objects of macro scales, cryo-EM reconstruction is particularly challenging. First, the images of particles are in the low signal-to-noise ratio (SNR) with unknown orientations. Such low SNR typically affects orientation estimation due to the severe corruption of the structural signal of particles. In addition, the flexible region of proteins induces conformational heterogeneity that disrupts orientation estimation and is harder to reconstruct. Conventional software packages (Scheres, 2012; Punjani et al., 2017) only reconstruct a small discrete set of conformations to reduce the complexity. However, such approaches often yield low-resolution reconstructions of flexible regions without guidance from human experts.

Recently, neural approaches exploit coordinate-based representations for heterogeneous cryo-EM reconstruction (Donnat et al., 2022; Zhong et al., 2021a;b; Levy et al., 2022b; Kimanius et al., 2022). To reduce the computational cost of image projection via the Fourier-slice theorem (Bracewell, 1956), they perform a 3D Fourier reconstruction. A downside, however, is that it is counter-intuitive to model local density changes between conformations in the Fourier domain. In contrast, 3DFlex (Punjani & Fleet, 2021) performs reconstruction in the real domain where motion is more naturally parameterized and more interpretable, but it requires an additional 3D canonical map as input.

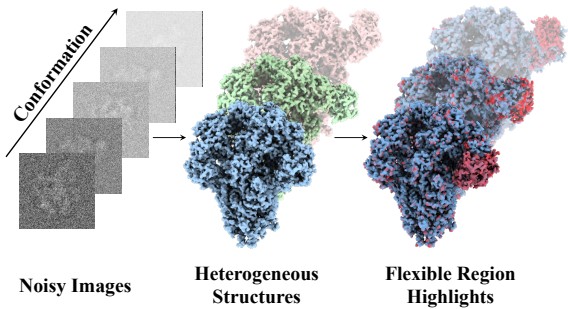

**Noisy Images**   **Heterogeneous Structures**   **Flexible Region Highlights**

Figure 1: **Overview.** With noisy images and pre-computed poses as inputs, our method continuously reconstructs the heterogeneous structures of proteins. It also enables the identification of local flexible motions through the analysis of 3D attention values.

In this paper, we propose *CryoFormer* for high-resolution continuous heterogeneous cryo-EM reconstruction (Fig. 1). Different from previous Fourier domain approaches (Zhong et al., 2021b; Levy et al., 2022a), CryoFormer is conducted in the **real** domain to facilitate the modeling of local flexible regions. Taking 2D particle images as inputs, our orientation encoder and deformation encoder first extract orientation representations and deformation features, respectively. Notably, to further disentangle orientation and conformation, we use pre-computed pose estimations to pre-train the orientation encoder. Next, we build an implicit feature volume in the real domain as the core of our approach to achieve higher resolution and recover continuous conformational states. Furthermore, we propose a novel query-based transformer decoder to obtain continuous heterogeneous density volume by integrating 3D spatial features with conformational features. The transformer-based decoder not only can model fine-grained structures but also supports highlighting spatial local changes for interpretability.

In addition, we present a new synthetic dataset of porcine epidemic diarrhea virus (PEDV) trimeric spike protein, which is a primary target for vaccine development and antigen analysis. Its dynamic movements from up to down in the domain 0 (D0) region modulate the enteric tropism of PEDV via binding to sialic acids (SAs) on the surface of enterocytes. We validate CryoFormer on the PEDV spike protein synthetic dataset and three existing datasets. Our approach outperforms the state-of-the-art methods including popular traditional software (Punjani et al., 2017; Punjani & Fleet, 2021) as well as recent neural approaches (Zhong et al., 2021a; Kimanius et al., 2022) in terms of spatial resolution on both synthetic and experimental datasets. Specifically, our method reveals dynamic regions of biological structures of the PEDV spike protein in our synthetic experiment, which implies functional areas but are hardly recovered by other methods. We will release our code and PEDV spike protein dataset.

## 2 RELATED WORK

**Conventional Cryo-EM Reconstruction.** Traditional cryo-EM reconstruction involves the creation of a low-resolution initial model (Leschziner & Nogales, 2006; Punjani et al., 2017) followed by the iterative refinement (Scheres, 2012; Punjani et al., 2017; Hohn et al., 2007). These algorithms perform reconstruction in the Fourier domain since this can reduce computational cost via Fourier-slice Theorem (Bracewell, 1956). When tackling structural heterogeneity, they classify conformational states into several discrete states (Scheres, 2010; Lyumkis et al., 2013). While this paradigm is sufficient when the structure has only a small number of discrete conformations, it is nearly impossible to individually reconstruct every state of a protein with continuous conformational changes in a flexible region (Plaschka et al., 2017).

**Dynamic Neural Representations.** Neural Radiance Fields (NeRFs) (Mildenhall et al., 2020) and their subsequent variants (Müller et al., 2022; Kerbl et al., 2023) have achieved impressive results in novel view synthesis. Numerous studies have introduced extensions of NeRF for dy-

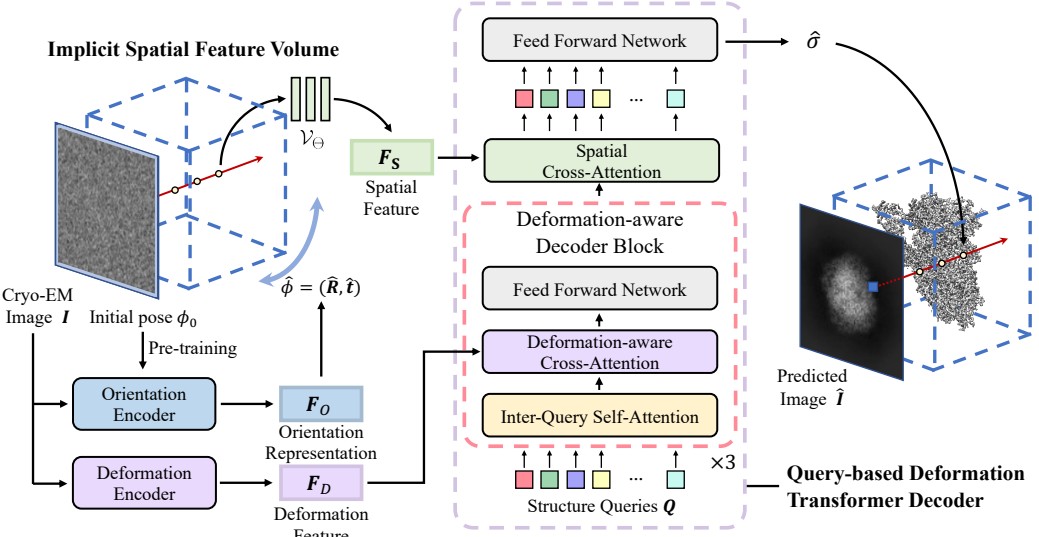

Figure 2: **Pipeline of CryoFormer. 1)** Given an input image, our orientation encoder and deformation encoder first extract orientation representations and deformation features. We use pre-computed pose estimations to pre-train the orientation encoder. **2)** We convert the orientation representation into a pose estimation and transformed coordinates are fed into our implicit neural spatial feature volume to produce a spatial feature. **3)** The spatial feature and the deformation image feature then interact in the deformation transformer decoder to output the density prediction.

namic scenes (Xian et al., 2021; Li et al., 2021; 2022; Park et al., 2021b; Yuan et al., 2021; Fang et al., 2022; Song et al., 2023). Most of these dynamic neural representations either construct a static canonical field and use a deformation field to warp this to the arbitrary timesteps (Pumarola et al., 2021; Tretschk et al., 2021; Zhang et al., 2021; Park et al., 2021a), or represent the scene using a 4D space-time grid representation, often with planar decomposition or hash functions for efficiency (Shao et al., 2023; Attal et al., 2023; Cao & Johnson, 2023; Fridovich-Keil et al., 2023).

**Neural Representations for Cryo-EM Reconstruction.** Recent work has widely adopted neural representations for cryo-EM reconstruction (Zhong et al., 2021a; Levy et al., 2022a;b; Shekarforoush et al., 2022). CryoDRGN (Zhong et al., 2021a) first proposed a VAE architecture to encode conformational states from images and decode it by an coordinated-based MLP that represents the 3D Fourier volume. Such a design can model continuous heterogeneity of protein and achieve higher spatial resolution compared with traditional methods. To reduce the computational cost of large MLPs, Kimanius et al. (2022) uses a voxel grid representation. To enable an end-to-end reconstruction, there are some *ab*-initio neural methods (Zhong et al., 2021b; Levy et al., 2022a;b; Chen et al., 2023) directly reconstruct protein from images without requiring pre-computed poses from traditional methods. CryoFIRE (Levy et al., 2022b) attempts to use an encoder to estimate poses from input image by minimizing reconstruction loss directly, but the performance is still limited due to the ambiguity of conformation and orientation in the extremely noisy image. To model the 3D local motion, Punjani & Fleet (2021) and Chen & Ludtke (2021) perform reconstruction in the real domain by using a flow field to model the structural motion, while they require a canonical structure as input.

## 3 METHOD

We propose CryoFormer, a novel approach that leverages a real domain implicit spatial feature volume coupled with a transformer-based network architecture for continuous heterogeneous cryo-EM reconstruction. In this section, we begin by laying out the cryo-EM image formation model in Sec. 3.1. We then introduce the procedural framework of CryoFormer (Fig. 2), encompassing orientation and deformation encoders (Sec. 3.2), an implicit spatial feature volume $\mathcal{V}_\Theta$ (Sec. 3.3) and a query-based deformation transformer (Sec. 3.4), with the training scheme described in Sec. 3.5

In this section, we use Attention to denote the scaled dot-product attention, which operates as

$$\text{Attention}(\boldsymbol{Q}, \boldsymbol{K}, \boldsymbol{V}) = \text{softmax}\left(\frac{\boldsymbol{Q}\boldsymbol{K}^T}{\sqrt{C}}\right)\boldsymbol{V}, \tag{1}$$

where $\boldsymbol{Q}, \boldsymbol{K}, \boldsymbol{V} \in \mathbb{R}^{N \times C}$ are called the query, key, and value matrices; $N$ and $C$ indicate the token number and the hidden dimension.

## 3.1 CRYO-EM IMAGE FORMATION MODEL

In the cryo-EM image formation model, the 3D biological structure is represented as a function $\sigma : \mathbb{R}^3 \mapsto \mathbb{R}^+$, which expresses the Coulomb potential induced by the atoms. To recover the potential function, the probing electron beam interacts with the electrostatic potential, resulting in projections $\{\mathbf{I}_i\}_{1 \leq i \leq n}$. Specifically, each projection can be expressed as

$$\mathbf{I}(x, y) = g \star \int_{\mathbb{R}} \sigma(\mathbf{R}^\top \mathbf{x} + \mathbf{t}) \, \mathrm{d}z + \epsilon, \quad \mathbf{x} = (x, y, z)^\top \tag{2}$$

where $\mathbf{R} \in SO(3)$ is an orientation representing the 3D rotation of the molecule and $\mathbf{t} = (t_x, t_y, 0)^\top$ is an in-plane translation corresponding to an offset between the center of projected particles and center of the image. The projection is, by convention, assumed to be along the $z$-direction after rotation. The image signal is convolved with $g$, a pre-estimated point spread function (PSF) for the microscope, before being corrupted with the noise $\epsilon$ and registered on a discrete grid of size $D \times D$, where $D$ is the size of the image along one dimension. We give a more detailed formulation for cryo-EM reconstruction in Sec. B.

## 3.2 IMAGE ENCODING FOR ORIENTATION AND DEFORMATION ESTIMATION

Given a set of input projections and their initial pose estimations, we extract latent representations for their orientation and conformational states using image encoders. Following (Zhong et al., 2021a; 2020) we adopt MLPs for both encoders.

**Orientation Encoding.** Given an input image $\mathbf{I}$, our orientation encoder predicts its orientation representation $\boldsymbol{F}_{\mathrm{O}} \in \mathbb{R}^8$. For optimization purposes, we represent rotations within the 6-dimensional space $\mathbb{S}^2 \times \mathbb{S}^2$ (Zhou et al., 2019) and translations with the remaining 2 dimensions. We map each image's orientation representation $\boldsymbol{F}_{\mathrm{O}}$ into a pose estimation $\hat{\phi} = (\hat{\mathbf{R}}, \hat{\mathbf{t}})$. In line with cry-oDRGN (Zhong et al., 2021a; 2020), for each image $\mathbf{I}$, we compute an initial pose estimation $\phi_0 = (\mathbf{R}_0, \mathbf{t}_0)$ (via off-the-shelf softwares (Scheres, 2012; Punjani et al., 2017)). While these initial estimations are not perfectly precise, particularly for cases with substantial motion, we utilize them as a guidance for our orientation encoder by pre-training it using

$$\mathcal{L}_{\mathrm{pose}} = \sum_{i=1}^{n} \left( \frac{1}{9} \left\| \hat{\mathbf{R}}_i - \mathbf{R}_{0,i} \right\|_2 + \frac{1}{2} \left\| \hat{\mathbf{t}}_i - \mathbf{t}_{0,i} \right\|_1 \right). \tag{3}$$

During the main stage of training, the orientation encoder estimates each image's pose to transform the 3D structure representation to minimize image loss (Eqn. 6). The gradient of the image loss is back-propagated to refine the pose encoder.

**Deformation Encoding.** The deformation encoder maps a projection $\mathbf{I}$ into a latent embedding for its conformational state, denoted as $\boldsymbol{F}_{\mathrm{D}}$. $\boldsymbol{F}_{\mathrm{D}}$ subsequently interacts with 3D spatial features within the query-based deformation transformer decoder to produce the density estimation $\hat{\sigma}$. In this way, our approach models the structural heterogeneity and produces the density estimation conditioned on the conformational state of each input image.

## 3.3 IMPLICIT SPATIAL FEATURE VOLUME IN THE REAL DOMAIN

In contrast with central slice sampling for Fourier domain reconstruction, real domain reconstruction requires sampling along $z$-direction for estimating each pixel. Consequently, leveraging a NeRF-like global coordinate-based MLP adopted by Zhong et al. (2021a) and Levy et al. (2022a;b) for real-domain cryo-EM reconstruction becomes computationally prohibitive. We instead adopt multi-resolution hash grid encoding (Müller et al., 2022) which has been used for real-time NeRF rendering as our 3D representation. We derive the high-dimensional spatial feature at each coordinate from it to better preserve the high-frequency details with highly reduced computational cost.

We use a hash grid $\mathcal{V}_\Theta$ parameterized by $\Theta$ as our 3D representation. For any given input coordinate $\mathbf{x} = (x, y, z)^\top$, the high-dimensional spatial feature is represented as

$$\boldsymbol{F}_S = \mathcal{V}_\Theta(\mathbf{x}; \Theta). \tag{4}$$



Figure 3: **Visualization of PEDV spike protein dataset.** On the left in each pair are our manually modified atomic models (PDB files) in their intermediate states; on the right are their corresponding converted density fields (MRC files).

This feature encapsulates the local structural information of the specified input location. It later interacts with deformation features $\boldsymbol{F}_\mathrm{D}$ in the deformation transformer decoder to yield the local density estimation for the coordinate conditioned on $\boldsymbol{F}_\mathrm{D}$.

### 3.4 QUERY-BASED DEFORMATION TRANSFORMER ARCHITECTURE

To generate the final density estimation $\hat{\sigma}$ at an arbitrary coordinate $\mathbf{x}$, we introduce a novel query-based deformation transformer decoder, where spatial features $\boldsymbol{F}_\mathrm{S}$ from the implicit feature volume interact with conformational state representation $\boldsymbol{F}_\mathrm{D}$. We denote randomly initialized learnable structure queries as $\boldsymbol{Q} \in \mathbb{R}^{N \times C}$ where $N$ is the number of queries and $C$ is the number of dimensions of each query. Spatial features and conformational states have shapes that match the dimensions of structure queries, specifically, $\boldsymbol{F}_\mathrm{S}, \boldsymbol{F}_\mathrm{D} \in \mathbb{R}^{N \times C}$.

**Deformation-aware Decoder Block.** Given an image with its conformational state $\boldsymbol{F}_\mathrm{D}$, the structure queries $\boldsymbol{Q}$ first interact with $\boldsymbol{F}_\mathrm{D}$ in the deformation-aware decoder blocks. Each deformation-aware block sequentially consists of an inter-query self-attention block ($\mathrm{Attention}(\boldsymbol{Q}, \boldsymbol{Q}, \boldsymbol{Q})$), a deformation-aware cross-attention layer, and a feed-forward network (FFN), where the deformation-aware cross-attention layer is computed as $\mathrm{Attention}(\boldsymbol{Q}, \boldsymbol{F}_\mathrm{D}, \boldsymbol{Q})$. We stack three decoder blocks for fusing deformation cues into structure queries.

**Spatial Density Estimation.** To estimate the density value at a specific coordinate, structure queries $\boldsymbol{Q}$ then interact with the spatial feature $\boldsymbol{F}_\mathrm{S}$ at this coordinate by spatial cross attention, computed as $\mathrm{Attention}(\boldsymbol{Q}, \boldsymbol{F}_\mathrm{S}, \boldsymbol{Q})$. Finally, an FFN maps the queries to the estimated density $\hat{\sigma}$.

### 3.5 TRAINING SCHEME

To train our system, we first calculate the projected pixel values as

$$\hat{\mathbf{I}}(x, y) = \hat{g} \star \int_{\mathbb{R}} \hat{\sigma}(\hat{\mathbf{R}}^\top \mathbf{x} + \hat{\mathbf{t}}) \, \mathrm{d}z + \epsilon, \quad \mathbf{x} = (x, y, z)^\top \tag{5}$$

where $\hat{g}$ is the point spread function (PSF) of the projected image, assumed to be known from contrast transfer function (CTF) correction (Rohou & Grigorieff, 2015) in the image pre-processing stage. The loss function for training is to measure the squared error between the observed images $\{\mathbf{I}_i\}_{1 \leq i \leq n}$ and the predicted images $\{\hat{\mathbf{I}}_i\}_{1 \leq i \leq n}$:

$$\mathcal{L} = \sum_{i=1}^{n} \left\| \mathbf{I}_i - \hat{\mathbf{I}}_i \right\|_2^2. \tag{6}$$

## 4 PEDV SPIKE PROTEIN DATASET

To evaluate CryoFormer and other heterogeneous cryo-EM reconstruction algorithms, we create a synthetic dataset of the spike protein of the *porcine epidemic diarrhea virus* (PEDV). The spike protein is a homotrimer, with each monomer containing a *domain 0* (D0) region that modulates the enteric tropism of PEDV by binding to *sialic acids* (SAs) on the surface of enterocytes (Hou et al., 2017) and can exist in both "up" and "down" states. Huang et al. (2022) determined the atomic coordinates and deposited them in the Protein Data Bank (PDB) (Berman et al., 2000) under the accession codes *7W6M* and *7W73*.

We utilized *Pymol* (DeLano et al., 2002) to manually supplement the reasonable process of the movement of the D0 region in the format of intermediate atomic models (Fig. 3 (a)). We converted

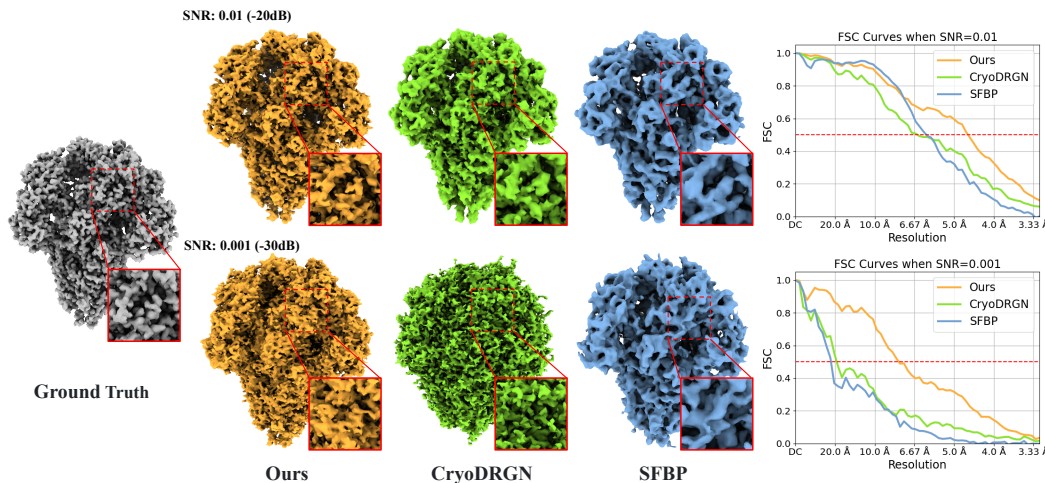

Figure 4: **Heterogenous reconstruction on PEDV spike dataset. Left:** Ground truth volume and reconstructed 3D volumes with SNR = 0.01 and SNR = 0.001. **Right:** Curves of the Fourier Shell Correlation (FSC) to the ground truth volumes. Our method produces a more refined reconstruction than both cryoDRGN and SFBP, especially in better recovery of the flexible D0 region across both noise scale levels. In addition, our approach yields the highest FSC curves.

these atomic models (PDB files) to discrete potential maps (MRC files) using *pdb2mrc* module from *EMAN2* (Tang et al., 2007), which were then projected into 2D images (Fig. 3 (b)). We then simulate the image formation model as in Eqn. 2 at uniformly sampled rotations and in-plane translations. On clean synthetic images, we add a zero-mean white Gaussian noise and apply the PSF. We adjust the noise scale to produce the desired SNR such as $0.1, 0.01$ and $0.001$. We will make the atomic models, density maps, and simulated projections publicly available.

# 5 RESULTS

In this section, we evaluate the performance of CryoFormer for homogeneous and heterogeneous cryo-EM reconstruction on two synthetic and two real experimental datasets and compare it with the state-of-the-art approaches. We also validate the effectiveness of our building components. Please also kindly refer to our appendix and supplementary video.

**Implementation Details** We adopt MLPs that contain 10 hidden layers of width 128 with ReLU activations for both the orientation encoder and the deformation encoder. For the implicit spatial feature volume, we utilized a hash grid with 16 levels, where the number of features in each level is 2, the hashmap size is $2^{15}$, and the base resolution is 16. This hash grid is followed by a tiny MLP with one layer and hidden dimension 16 to extract final spatial features. For the query-based deformation transformer, we adopt $N = 64$ queries with $C = 64$ dimensions. For synthetic datasets, we use ground truth poses for all the methods. For real datasets, we use CryoSPARC (Punjani et al., 2017) for initial pose estimation (following Zhong et al. (2021a) and Kimanius et al. (2022)). All experiments including training and testing have been conducted on a single NVIDIA GeForce RTX 3090 Ti GPU.

**Reconstruction Metrics.** For quantitative evaluations, we employ the Fourier Shell Correlation (FSC) curves, defined as the frequency correlation between two density maps (Harauz & van Heel, 1986). A higher FSC curve indicates a better reconstruction result. For synthetic datasets, we compute FSC between the reconstructions and the corresponding ground truths and take the average if there are multiple conformational states. For real experimental datasets where the ground truth volume is unavailable, we compute FSC between two half-maps, each reconstructed from half the particle dataset. For EMPIAR-10180 (real data with multiple states), we conduct a principal component analysis on the deformation latent space obtained from both reconstructions. By uniform sampling along the principal component axes, we obtain corresponding volumes at the same conformational states and report the average of 10 FSC curves. We report the spatial resolutions of the reconstructed volumes, defined as the inverse of the maximum frequency at which the FSC exceeds a threshold ($0.5$ for synthetic datasets and $0.143$ for experimental datasets).

## 5.1 HETEROGENEOUS RECONSTRUCTION ON SYNTHETIC DATASETS

We first evaluate CryoFormer on two synthetic datasets: 1) the synthetic dataset proposed by (Zhong et al., 2021a), generated from an atomic model of a protein complex (cryoDRGN synthetic dataset), and 2) our proposed PEDV spike dataset. We compare CryoFormer against **cryoDRGN** (Zhong et al., 2021a) (MLPs) and Sparse Fourier Backpropagation (**SFBP**) (Kimanius et al., 2022) (voxel grids) as representatives of coordinate-based methods.

**CryoDRGN Synthetic Dataset.** This dataset contains $50,000$ images with size $D = 128$ (pixel size $= 1.0$Å) and SNR $= 0.1(-10$dB$)$ from an atomic model of a protein complex containing a 1D continuous motion (Zhong et al., 2021a). We demonstrate that our reconstruction aligns quantitatively with the ground truth in Tab.1. Detailed results on the cryoDRGN synthetic dataset can be found in Sec. E.1.

**PEDV Spike Protein Dataset.** To further investigate CryoFormer's capability, we use the new PEDV spike protein dataset containing $50,000$ image with size $D = 128$ (pixel size $= 1.6$Å) to create more challenging experiment settings. These settings involve more complicated 3D density maps and lower signal-to-noise ratios (SNR). We conducted experiments in two different levels of noise scale: SNR $= 0.01(-20$dB$)$ and SNR $= 0.001(-30$dB$)$. As is shown in Fig. 4 (left panel), our method produces a more refined reconstruction than cryoDRGN and SFBP, with a better-recovered flexible D0 region under both levels of the noise scale. In Fig. 5, we jointly visualize the reconstructed states from various approaches with SNR $= 0.1$ $(-10$dB$)$. Our results not only

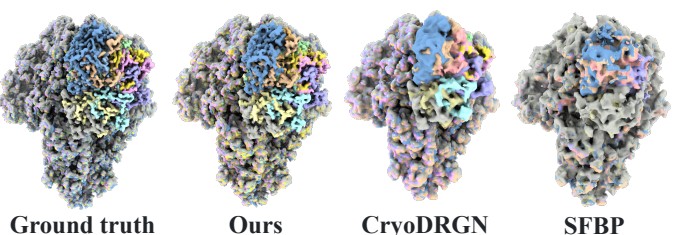

**Ground truth**    **Ours**    **CryoDRGN**    **SFBP**

Figure 5: **Qualitative comparison of the joint visualizations of multiple reconstructed states on PEDV spike dataset.** Our approach recovers all the conformational states and captures fine-grained details. In contrast, the baselines either exhibit lower spatial resolution or fail to capture all the states.

|  | CryoDRGN Synthetic | PEDV (SNR=0.01) | PEDV (SNR=0.001) |
|---|---|---|---|
| CryoDRGN | 3.45 | 6.5 | 19.21 |
| SFBP | 2.18 | 6.06 | 20.8 |
| Ours | **2.03** | **4.6** | **7.47** |

Table 1: **Quantitative comparison for heterogeneous reconstruction on synthetic datasets.** Spatial resolution (in Å, ↓) is quantified by an FSC=0.5 threshold.

recover all the conformational states but also capture fine-grained details. In contrast, cryoDRGN's reconstructions exhibit lower spatial resolutions for details, and SFBP fails to capture all the states. The quantitative results from Fig.4 (right panel) and Tab.1 indicate that our reconstruction outperforms competing approaches in terms of the FSC curve and the spatial resolution, with an exception in a low-frequency region where our curve marginally falls below that of SFBP.

## 5.2 HOMOGENEOUS RECONSTRUCTION ON EXPERIMENTAL DATASETS

To demonstrate CryoFormer's reconstruction performance on the real experimental data, we begin with homogeneous reconstruction on an experimental dataset with the ignorable biological motions from EMPIAR-10028 (Wong et al., 2014), consisting of 105,247 images of the 80S ribosome downsampled to $D = 256$ (pixel size $= 1.88$Å). Our baselines include neural reconstruction approaches **cryoDRGN** (Zhong et al., 2021a) and **SFBP** (Kimanius et al., 2022) as well as a traditional state-of-the-art method **cryoSPARC** (Punjani et al., 2017). As illustrated in the left panel of Fig. 6, our method manages to recover the shape and integrity of detailed structures like the $\alpha$-helices (as seen in the zoom-in region) in contrast to baseline approaches. The right panel of Fig. 6 shows that our FSC curve consistently surpasses those of all the baselines, quantitatively demonstrating the accuracy of our reconstructed details. For the resolution, defined as the inverse of the maximum frequency at which the FSC exceeds $0.143$, our approach achieves the theoretical maximum value of 3.80Å. This surpasses CryoDRGN (3.93Å), SFBP (6.19Å), and CryoSPARC (8.63Å).

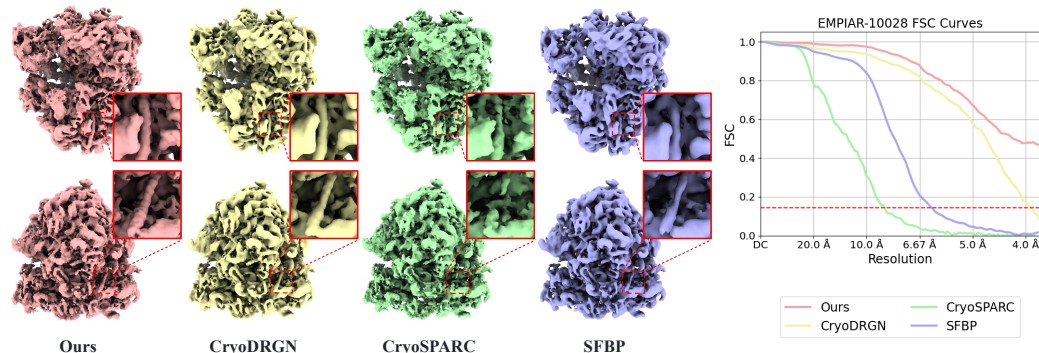

Figure 6: **Homogeneous cryo-EM reconstruction on EMPIAR-10028. Left:** Reconstructed 3D volumes. **Right:** Curves of FSC between half-maps. Our method recovers detailed structures, such as the $\alpha$-helices in zoom-in regions, more clearly than baselines and achieves the highest FSC curve.

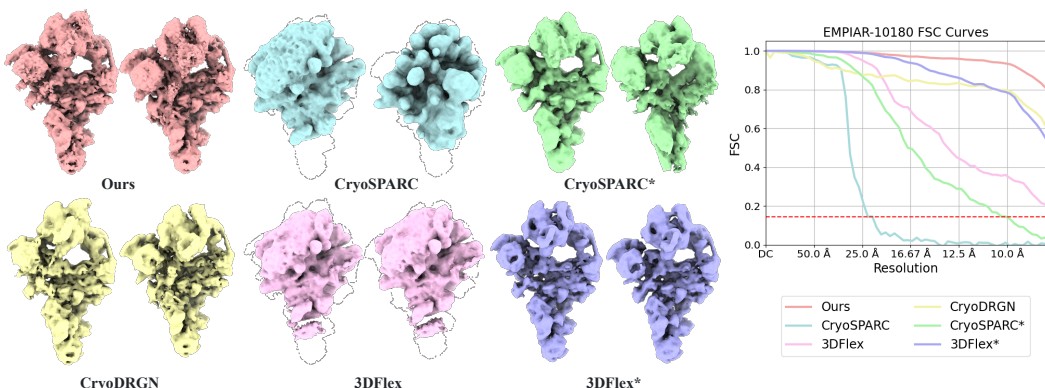

Figure 7: **Heterogeneous cryo-EM reconstruction on EMPIAR-10180. Left:** Reconstructed 3D volumes. We display two states for each method. **Right:** Curves of FSC between two half-maps. Our method manages to recover continuous motions with a clearer outline of the secondary structure and achieve the highest FSC curve.

### 5.3 HETEROGENEOUS RECONSTRUCTION ON EXPERIMENTAL DATASETS

To test CryoFormer's capability of heterogeneous reconstruction on real experimental datasets, we evaluate it on EMPIAR-10180 (Plaschka et al., 2017), consisting of 327,490 images of a pre-catalytic spliceosome downsampled to $D = 128$ (pixel size = 4.2475Å). We compare CryoFormer with **CryoDRGN** (Zhong et al., 2021a), **CryoSPARC** (Punjani et al., 2017) and **3DFlex**. Notably, we follow the original paper of 3DFlex to use CryoSPARC's reconstructed volume as an input canonical volume reference. As low-quality particles highly decrease CryoSPARC's reconstruction performance, we manually remove particles with lower quality after 2D classification to improve subsequent reconstruction performance and denote this result as **CryoSPARC\***. In addition, we denote 3DFlex with CryoSPARC\*'s reconstruction as the input canonical reference map as **3DFlex\***. As shown on the left side of Fig. 7, CryoSPARC and thus 3DFlex fail to provide reasonable reconstructions. Our method and CryoDRGN, CryoSPARC\*, and 3DFlex\* manage to maintain structural integrity during dynamic processes, while our reconstructions exhibit a clear outline of the secondary structure. Quantitatively, as depicted on the right side of Fig. 7, our method achieves the highest FSC curve.

### 5.4 EVALUATION

To validate our architecture designs of CryoFormer, we conduct the following evaluations on our synthetic PEDV spike protein dataset. We generate a dataset with 50,000 projections of the ground truth volume with SNR = 0.1. We sample particle rotations uniformly from $SO(3)$ space and particle in-plane translations uniformly from $[-10\text{pix.}, 10\text{pix.}]^2$ space. To simulate imperfect pre-computed poses, we perturb the ground truth rotations using additive noise ($\mathcal{N}(\mathbf{0}, 0.1\mathbf{I})$), and the translations using another uniform distribution $[-10\text{pix.}, 10\text{pix.}]^2$. We use the perturbed poses as simulated initial coarse estimations for the pre-training of our orientation encoder. We report the

| Ori. Encoder Refinement | Deformation Encoder | Spatial Cross-Attention | Domain | Resolution(↓) | Rot. Error(↓) | Trans. Error(↓) |
|---|---|---|---|---|---|---|
| | ✓ | ✓ | Real | 26.1 | 0.144 | 0.138 |
| ✓ | | ✓ | Real | 8.7 | 0.118 | 0.128 |
| ✓ | ✓ | | Real | 6.5 | 0.066 | 0.036 |
| ✓ | ✓ | ✓ | Fourier | 7.3 | 0.088 | 0.021 |
| ✓ | ✓ | ✓ | Real | **4.1** | **0.030** | **0.018** |

Table 2: **Quantitative ablation study.** Resolution is reported using the FSC = 0.5 criterion, in Å. Rotation error is the mean square Frobenius norm between predicted and ground truth. Translation error is the mean L2-norm over the image side length.

resolution at the 0.5 cutoff in Å and the errors of the final pose estimations. We conduct analysis on the orientation encoder and the deformation encoder in Sec. E.2.

**Orientation Encoder Refinement.** We test a variant of our method without fine-tuning the orientation encoder using image loss. As seen in Tab. 2, the refined orientation encoder fixes inaccurate initial estimations, and without refinement, the model cannot reconstruct structures reasonably.

**Deformation Encoder.** We conducted experiments with CryoFormer without the deformation encoder for extracting deformation features. As evident from Tab. 2, this variant cannot accurately account for structural motions, resulting in a lower resolution.

**Real Domain Representation.** We conducted experiments using the Fourier domain variant of our method. As evident from Tab. 2, real domain reconstruction achieves significantly better resolutions and lower pose errors.

**Query-based Deformation Transformer Decoder.** To verify the effectiveness of our query-based deformation transformer decoder, we experimented with a variant of our method, replacing it with simple concatenation and an MLP. Tab. 2 shows that the replacement will decrease CryoFormer's performance. Furthermore, we can analyze attention maps to locate flexible regions. The 3D attention maps are computed at each coordinate through spatial cross-attention between its spatial feature and deformation-aware queries. For visualization, we map the attention value of one channel to the surface color of the reconstructed volume. As shown in Fig. 8, the displayed channel of attention map reflects a flexible region of the PEDV spike.

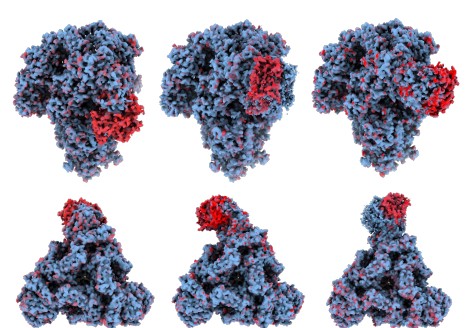

Figure 8: **Visualization of PEDV spike protein attention map.** We map the attention value to the surface color of the reconstructed volume of PEDV spike protein. The highlight (high attention value) reflects its flexible regions. Every row shows three different states from the same perspective.

## 6 DISCUSSION

**Limitations.** As the first trial to achieve continuous heterogeneous reconstruction of protein in real space without the need for a 3D reference map, CryoFormer still suffers from some limitations. Though we use hash encoding for efficient training and inferring, our method still demands a significant amount of computational resources and requires a long training time for full-resolution reconstruction due to we have to query every voxel in implicit feature volume for 3D projection operation, as discussed in Sec. E.2. Also, our orientation encoder depends on pre-training with initial pose estimation so CryoFormer cannot handle *ab*-initio reconstruction while CryoFormer shows the capability to further refine them during the training.

**Conclusion.** We have introduced CryoFormer for high-resolution continuous heterogeneous cryo-EM reconstruction. Our approach builds an implicit feature volume directly in the real domain as the 3D representation to facilitate the modeling of local flexible regions. Furthermore, we propose a novel query-based deformation transformer decoder to enhance the quality of reconstruction. Our approach can refine pre-computed pose estimations and locate flexible regions. Quantitative and qualitative experiment results show that our approach outperforms traditional methods and recent neural methods on both real datasets and synthetic datasets. In the future, we believe real-domain neural reconstruction methods can play a greater role in cryo-EM applications.

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

APPENDIX

## A VIDEO

For better visualization of cryo-EM reconstruction results, we use ChimeraX (Goddard et al., 2018) to create a set of video visualizations with free-viewpoint rendering and multiple conformational states. Please refer to *supplementary_video.mp4* for more results and evaluations of CryoFormer.

## B IMAGING MODEL OF CRYO-EM

Cryo-EM is a revolutionary imaging technique used to discover the 3D structure of biomolecules, including proteins and viruses. In a typical cryo-EM experiment, a purified sample containing many instances of the specimen is plunged into a cryogenic liquid, such as liquid ethane. This causes the molecules to freeze within a vitreous ice matrix. The frozen sample is loaded into a transmission electron microscope and exposed to parallel electron beams (Fig. A (a)), resulting in projections of the Coulomb scattering potential of the molecules (Fig. A (b)). These raw micrographs can then be processed by algorithms to reconstruct the volume.

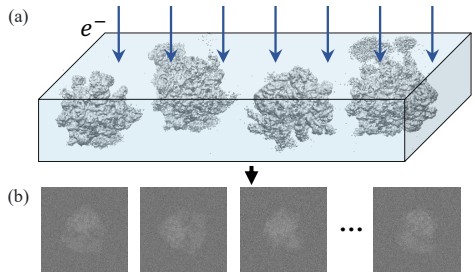

Figure A: **Illustration of a cryo-EM experiment.** (a) A sample containing molecules of interest is frozen in a thin layer of vitreous ice and exposed to parallel electron beams. (b) Two-dimensional projection images of the Coulomb scattering potential of the molecules.

In the cryo-EM image formation model, the 3D biological structure is represented as a function $\sigma : \mathbb{R}^3 \mapsto \mathbb{R}^+$, which expresses the Coulomb potential induced by the atoms. The probing electron beam interacts with the electrostatic potential, and ideally, one can formulate its clean projections without any corruption as

$$\mathbf{I}_{\text{clean}}(x, y) = \int_{\mathbb{R}} \sigma(\mathbf{R}^\top \mathbf{x} + \mathbf{t}) \, \mathrm{d}z, \quad \mathbf{x} = (x, y, z)^\top, \tag{7}$$

where $\mathbf{R} \in SO(3)$ is an orientation representing the 3D rotation of the molecule and $\mathbf{t} = (t_x, t_y, 0)^\top$ is an in-plane translation corresponding to an offset between the center of projected particles and center of the image. The projection is, by convention, assumed to be along the $z$-direction after rotation. In practice, images are intentionally captured under defocus in order to improve the contrast, which is modeled by convolving the clean images with a point spread function (PSF) $g$. All sources of noise are modeled with an additional term $\epsilon$ and are usually assumed to be Gaussian white noise. The image formation model considering PSF and noise is expressed as

$$\mathbf{I} = g \star \mathbf{I}_{\text{clean}} + \epsilon. \tag{8}$$

## C STRUCTURAL DEFORMATIONS IN DIFFERENT DOMAINS

To elucidate the advantages of reconstructing motions in the real domain, we use two neighbor conformational states of the PEDV spike protein as an illustration. As shown in Fig. B, there are two volumes colored in grey and yellow that represent state 0 and state 1, respectively, the difference between these two states is manifested as a local motion in the real domain. However, in the Fourier domain, they present a global difference.

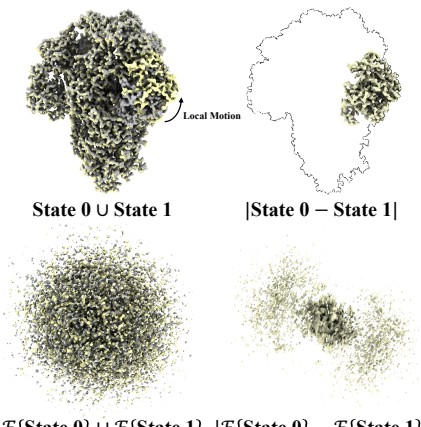

**State 0 ∪ State 1**     **|State 0 − State 1|**

$\mathcal{F}\{\text{State 0}\} \cup \mathcal{F}\{\text{State 1}\}$   $|\mathcal{F}\{\text{State 0}\} - \mathcal{F}\{\text{State 1}\}|$

Figure B: Visualization of two conformational states and their differences in the real domain and the Fourier domain.

Fourier reconstruction methods such as CryoDRGN (Zhong et al., 2021a), require the decoder to model the global and large value changes in the Fourier domain between two neighbor states, which should have very similar conformational embedding from the image encoder. However, our approach performs reconstruction in real domain so the neural representation only needs to model local and small changes between two neighbor conformational states.

# D  EXPERIMENT DETAILS

## D.1  DATASETS

We adopt a number of synthetic and real experimental datasets in our experiments. We show sample images in Fig. C and list parameters in Tab. A.

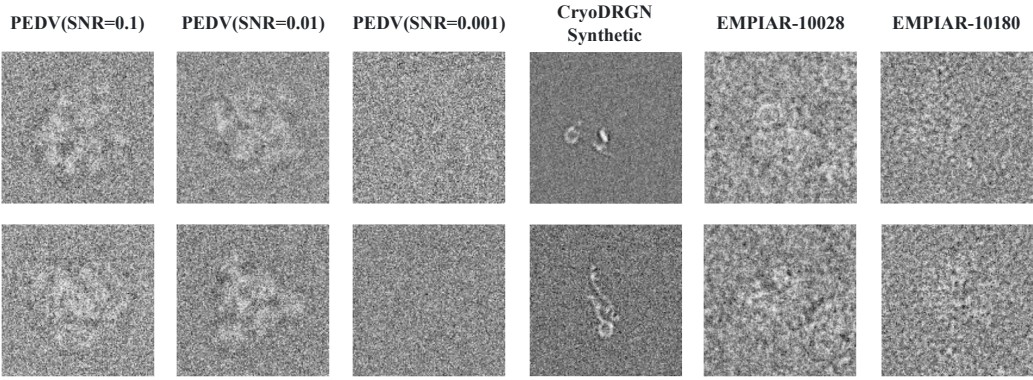

Figure C: **Sample images from synthetic and experimental datasets.**

## D.2  BASELINE SETTINGS

**CryoDRGN (Zhong et al., 2021a).** We use the official repository[1]. The number of latent dimensions is 8, and both the encoder and decoder have 3 layers and 1024 units per layer.

---

[1]The repository is available at `https://github.com/zhonge/cryodrgn`.

| Dataset | Number of Particles | Image Resolution (pixel) | Pixel Size (Å) | Number of States |
|---|---|---|---|---|
| PEDV Spike Protein | 50,000 | 128 | 1.60 | 10 |
| CryoDRGN Synthetic | 50,000 | 128 | 1.00 | 10 |
| EMPIAR-10028 | 105,247 | 256 | 1.88 | N/A |
| EMPIAR-10180 | 327,490 | 128 | 4.25 | N/A |

Table A: **Summary of the parameters for synthetic and experimental datasets.**

**Sparse Fourier Backpropagation (Kimanius et al., 2022).** As there is no open-source code available for SFBP, we re-implement the method, following the same setting as in the original paper, with an encoder consisting of five layers and a decoder consisting of a single linear layer, 256 units per layer. The number of structural bases is 16.

**CryoSPARC (Punjani et al., 2017) and 3DFlex (Punjani & Fleet, 2021).** We use CryoSPARC v4.2.1. For homogenerous dataset, we follow the typical workflow (import particle stacks, perform *ab*-initio reconstruction before homogeneous refinement) with default parameters. For heterogeneous dataset, we change the number of classes parameter in *ab*-initio reconstruction job to 5, and run heterogenous refinement between *ab*-initio reconstruction and homogeneous refinement. We run 3DFlex following the official tutorial with default parameters using CryoSPARC's reconstruction map as input canonical maps.

# E    EXPERIMENTAL RESULTS (CONT'D)

## E.1    CRYODGRN SYNTHETIC DATASET RESULT

In Fig.D (left and middle panels), we demonstrate that our reconstruction not only aligns qualitatively with the ground truth but is also adept at capturing the dynamics of the structure. Notably, although we only illustrate 10 structures sampled from various points along the latent space, our approach is capable of reconstructing the full continuous conformational states. As shown in Fig. D (right panel), our reconstruction's FSC curve is predominantly higher than that of the baselines, with a singular exception where it falls marginally below cryoDRGN in a low-frequency region.

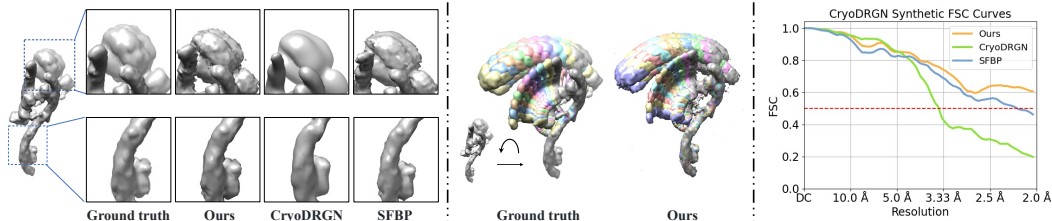

Figure D: **Heterogenous reconstruction on the cryoDRGN synthetic dataset. Left:** The ground truth volume and reconstructions from our approach and baselines. **Middle:** Multiple conformational states of the ground truth and our reconstruction. **Right:** Curves of the Fourier Shell Correlation (FSC) to the ground truth volumes. Our reconstruction qualitatively aligns with the ground truth, with an FSC curve predominantly higher than baselines'.

## E.2    ANALYSIS

To better understand the behaviors of our approach and its building components, we conduct the following analysis on the synthesized datasets from the PEDV spike proteins. In this section, we follow the same experimental setting as Sec. 5.4.

**Deformation Encoder.** To analyze our deformation encoder, we perform principal component analysis (PCA) on the deformation features of all images and plot their distributions in Fig. E. We apply K-means algorithm with 10 clusters and color-code the points with their corresponding K-means labels. We display the reconstructed volumes corresponding to 6 of the cluster centers in Fig. F.

**Orientation Encoder Refinement.**    To analyze our orientation encoder, we visualized the initial pose estimation and the refined pose in Fig.G (left panel) and the video. It can be seen that after the

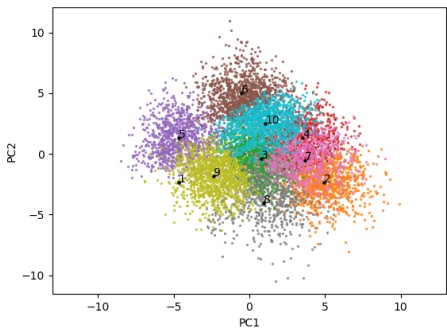

Figure E: **Distribution of deformation features.** We visualize the distribution of all the images' deformation features in 2D with PCA and color-code the points by their corresponding K-means labels. The cluster centers are annotated.

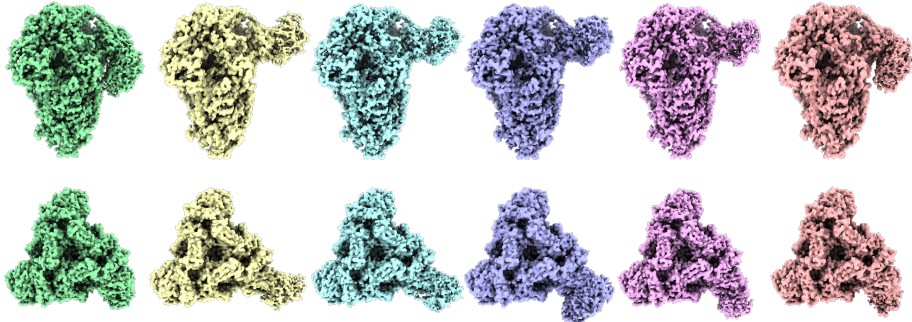

Figure F: **Reconstructed states of PEDV spike protein.** We sample six cluster centers in Fig. E and visualize the corresponding reconstructed states. The volumes, from left to right, correspond to the following clusters: 5, 6, 4, 2, 8, and 1.

refinement of the orientation encoder via image loss, the previously inaccurate initial pose estimation has been optimized and aligns with the ground truth. In Fig.G (right panel), we compare multiple reconstructed states of the variant without the refinement approach and our full model. It can be observed that if we do not refine the pose estimation, the initial inaccurate pose estimation results in a very low resolution of the reconstruction.

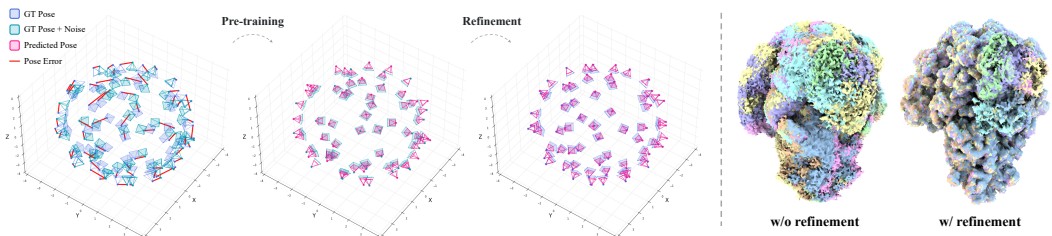

Figure G: **Left:** Visualization of the initial poses and the predicted poses from orientation encoder after pre-training and refinement. We draw the predicted pose with a smaller size for better distinction. **Right:** Reconstructed volumes without and with the refinement of the orientation encoder.

**Running Time.** To demonstrate the runtime of our algorithm, we present the execution times on a single NVIDIA GeForce RTX 3090 Ti GPU in Tab. B. We also report the runtime of CryoDRGN as a reference. The reported time corresponds to 20 epochs. The timings for our method include both the pre-training of the orientation encoder (200 epochs) and the training of the system (20 epochs).

As can be observed, when the image size is 128, our algorithm and CryoDRGN have comparable runtimes. However, for a larger image size of 256, our approach takes significantly more time. This may be attributed to the increased time complexity of the spatial cross-attention mechanism in our method when processing high-resolution images.

| Dataset | Ours | CryoDRGN |
|---|---|---|
| CryoDRGN Synthetic | 2h22min | 2h48min |
| PEDV Spike Protein | 3h12min | 3h16min |
| EMPIAR-10028 | 31h14min | 8h12min |
| EMPIAR-10180 | 18h34min | 19h50min |

Table B: **Comparison of processing times between our method and CryoDRGN.**

