# OpenReview forum: "CryoFormer: Continuous Heterogeneous Cryo-EM Reconstruction using Transformer-based Neural Representations"
_ICLR.cc/2024/Conference — ICLR 2024 Conference Withdrawn Submission_

### Official Review · Reviewer_cxj8 · 2023-10-30

**Soundness:** 3 good
**Presentation:** 4 excellent
**Contribution:** 3 good
**Rating:** 3
**Confidence:** 4

**Summary:**

The authors attempt to model cryo-EM density in real space to better capture heterogeneity. They introduce a novel query-based deformation transformer decoder to improve reconstruction quality. Experimental results prove the effectiveness of the approach.

**Strengths:**

- The authors conduct experiments on four datasets, including two synthetic datasets and two experimental datasets. The experiments are comprehensive.

- The manuscript is well organized and the writing is easy to follow.

- The motivation for modeling in real space makes sense to me. Experimental results demonstrate the effectiveness of the approach.

**Weaknesses:**

- Evaulating the decoder is computationally heavy in real space. See questions.

- The motivation for incorporating the Transformer architecture is unclear. Transformer is often used to model interactions between different parts, such as patches in CV and tokens in NLP. In this study, Transformer is more like another feature extractor, which may be stronger than vanilla MLP.

  - What do query embeddings mean?

  - How do you choose the value of N and C?

  - You mentioned that you tried to replace the Transformer with MLP. What is the Transformer/MLP's parameter number?

- More ablation studies on the experimental dataset are required to show the need for each component.

  - In the homogenous reconstruction experiment (Figure 6, EMPIAR 10028), the model performs slightly better than CryoDRGN. Does this mean that higher resolution results from "modeling in real space", rather than "so many deformation components"?

  - It is better to test which component contributes the most to the performance gain on the real dataset. Does it come from finetuning the orientation encoder? Does it come from modeling in real space?

  - How does the performance change against the number and channel of query embeddings?

- The 3DFlex's result in Figure 7 has some conflict with my personal experience. I have tested 3DFlex against EMPIAR-10180 and its performance is not as bad as you illustrated in Figure 7.

**Questions:**

- In Appendix E, you mention that "However, for a larger image size of 256, our approach takes significantly more time." What is the running time?

- Can you provide a baseline of:

  - a cryoDRGN-style network that models density in real space

  - a cryoDRGN-style network that models density in real space, with pose refinement

- Can the model be used to sample in the latent space?

---

> ### Author Response · Authors · 2023-11-14
>
> We briefly clarify that we have also provided a refined result for 3DFlex by manually removing particles of low quality after 2D classification in Figure 7 for a fair comparison.

---

### Official Review · Reviewer_LTYj · 2023-10-30

**Soundness:** 2 fair
**Presentation:** 3 good
**Contribution:** 2 fair
**Rating:** 5
**Confidence:** 5

**Summary:**

The authors propose a new machine learning algorithm for heterogeneous cryo-EM reconstruction in real space with vision transformers.

**Strengths:**

- This is the first application of vision transformers to cryo-EM reconstruction
- The method as a whole is novel and the results seem competitive
- This paper contains a nice ablation study that makes it easier to appreciate the relevance of different design decisions.

**Weaknesses:**

Major weaknesses:
- The authors have included a number of baseline comparisons. However, for one of these approaches (Kimanius et al., 2022), the authors have claimed that an implementation for the method does not exist and that they have reimplemented this approach themselves (pg 16). As the results for this method may be affected by subtle differences in the implementation, I **strongly** suggest the authors use the open-source implementation of the paper, which is linked in the methods section of Kimanius et al., 2022 (pg 4).
- The reconstruction metrics used (apart from the synthetic data case) are very unclear. It should be explained in the main text of the paper what volumes are being compared for the FSC and whether the full pipeline is independent for the half-sets, or if it is only the decoder. Proper metrics for heterogeneous cryo-EM reconstruction that are only computational and are separated from the underlying biology are lacking and a research question in itself. How exactly these are computed has major ramifications for whether the results of this method are improving over the baselines.
- It is not clear how important the main contributions of this paper (deformation decoder etc) are versus the improved poses obtained through the framework. The authors should re-run the baselines with the poses that they obtained from their pipeline and mention if that improves results for SFBP and cryoDRGN.

Minor points:
- At the end of page 1, the authors claim that conventional cryo-EM reconstruction algorithms pick a limited number of classes in 3D classification just to improve the computational burden of the calculation. However, this is not the case. The real bottleneck here is that the more classes that are used, the fewer particles one has for each class. Thus the reconstruction quality is harmed. Please see Nakane et al., 2018.
- On page 6 in implementation details, it is claimed that SFBP uses cryoSPARC for the initial poses, but it uses RELION.



Citations:
- Takanori Nakane, Dari Kimanius, Erik Lindahl, Sjors HW Scheres (2018) Characterisation of molecular motions in cryo-EM single-particle data by multi-body refinement in RELION. eLife
- Dari Kimanius, Kiarash Jamali, Sjors HW Scheres (2022) Sparse fourier backpropagation in cryo-em reconstruction. Advances in Neural Information Processing Systems(NeurIPS).

**Questions:**

- What is the average change in rotation and translations that the pose encoder predicts compared to the initial poses?
- How important is it to have an encoder for the poses instead of doing gradient descent directly on the poses?
- For equation 3 (the loss function of the pose encoder), did the authors try any other formulations? Perhaps directly in the output space of their network in $\mathbb{S}^2 \times \mathbb{S}^2 \times \mathbb{R}^2$? Were there any optimization issues with this loss?
- How stable are the pose estimates optimized through this pipeline? I.e. if you save the poses predicted at the end of the reconstruction and then use these as the starting point for another reconstruction with reinitialized weights, do the poses remain stable or do they still change?

---

> ### Author Response · Authors · 2023-11-14
>
> We regret our oversight in initially missing the fact that the authors had already released the code for their method. We have explored the provided repository, noting that it appears to be in development with incomplete instructions. We will update our experimental results, replacing those from SFBP if possible.

---

### Official Review · Reviewer_TQAh · 2023-10-30

**Soundness:** 3 good
**Presentation:** 3 good
**Contribution:** 3 good
**Rating:** 5
**Confidence:** 3

**Summary:**

Reconstructing 3D structures of macromolecules from 2D Cryo-EM images is a significant problem while challenging. This paper proposes a novel approach, CryoFormer, for continuous heterogeneous cryo-EM reconstruction. Their approach introduces a deformation transformer decoder to improve the reconstruction quality and is able to locate flexible regions. Their method outperforms baseline methods on both synthetic and real public dataset.

**Strengths:**

• Originality: While NERF has been around for a while and the idea of using learning-based method for cryo-EM reconstruction is not new, this paper address the limitation of accurately capturing local flexible regions. They do so by modeling the local flexible regions in real-domain.
    • Significance: A well-reconstructed set of CryoEM data is essential for understanding properties of macromolecules through other downstream tasks such as image segmentation and classifications. This paper demonstrates good performance of reconstruction of the Cryo EM data compared to other SOTA methods.
    • Quality: Good explanation of their methods and well discussion of the results on both synthetic and real dataset.
    • New dataset generation and public available

**Weaknesses:**

• The submission only compares their approach against two of other current approaches: CryoDRGN and SFBP. There are also other deep learning based 3D reconstruction algorithms have been proposed over the past few years that the paper should discuss and compare. For example “Isotropic reconstruction for electron tomography with deep learning” Zhou et al
    • More discussion and results of time/computing resources consumption of reconstruction with proposed method and other methods should be added.
    • With the improved local spatial features reconstruction, the down-streaming tasks such as cryoEM segmentation and classification should also be improved. Some discussion/experiment can be added.

**Questions:**

• What are the fundamental differences between your methods and other reconstruction methods such as Isotropic reconstruction for electron tomography with deep learning” Zhou et al
    • How much does it take to perform reconstruction? What is the computing resource used. How are they compared against your baseline?
    • With improved fine-grained details reconstruction, do you expect to see improvement on downstream tasks such as CryoEM image classification/segmentation. Have you tried compare the performance using the original dataset and reconstructed dataset?

---

> ### Author Response · Authors · 2023-11-14
>
> It appears there may be a misunderstanding regarding the relationship between single-particle cryo-electron microscopy (cryo-EM) and cryo-electron tomography (cryo-ET) experiments. IsoNet, the baseline suggested, is specifically designed for cryo-ET scenarios where the specimen's limited rotation range leads to the 'missing wedge' artifacts in reconstructed volumes. IsoNet is a learning-based method developed to address this particular artifact. Consequently, it may not be an appropriate baseline for single-particle cryo-EM reconstruction tasks, as these two techniques address different challenges and aspects of electron microscopy.

---

### Official Review · Reviewer_96CH · 2023-10-31

**Soundness:** 3 good
**Presentation:** 3 good
**Contribution:** 3 good
**Rating:** 6
**Confidence:** 2

**Summary:**

The paper proposed CryoFormer for high-resolution continuous heterogeneous cryo-EM reconstruction. It builds an implicit feature volume directly in the real domain as the 3D representation to facilitate the modeling of local flexible regions. It also proposes a query-based deformation transformer decoder to enhance the quality of reconstruction. The approach can refine pre-computed pose estimations and locate flexible regions. Experimental results show the proposed approach outperforms traditional methods and recent neural methods on both real datasets and synthetic datasets.

**Strengths:**

The idea of using an implicit feature volume is good and has merit. The query-based transformer decoder to obtain continuous heterogeneous density volume by integrating 3D spatial features with conformational features seems to work well and can model fine-grained structures but also supports highlighting spatial local changes for interpretability. The paper also creates a new synthetic dataset of porcine epidemic diarrhea virus (PEDV) trimeric spike protein,  a primary target for vaccine development and antigen analysis. The dataset would be very useful.

**Weaknesses:**

I would like to see more ablation studies if possible.

**Questions:**

No.

---

### Author Response · Authors · 2023-11-14

We extend our sincere thanks to all the reviewers for their insightful and constructive comments. To fully incorporate the feedback and enhance the quality of our work, we have decided to withdraw our submission from this venue.